# FROM FOURIER TO NEURAL ODEs: FLOW MATCHING FOR MODELING COMPLEX SYSTEMS

## ABSTRACT

Modeling complex systems using standard neural ordinary differential equations (NODEs) often faces some essential challenges, including high computational costs and susceptibility to local optima. To address these challenges, we propose a simulation-free framework, called Fourier NODEs (FNODEs), that effectively trains NODEs by directly matching the target vector field based on Fourier analysis. Specifically, we employ the Fourier analysis to estimate temporal and potential high-order spatial gradients from noisy observational data. We then incorporate the estimated spatial gradients as additional inputs to a neural network. Furthermore, we utilize the estimated temporal gradient as the optimization objective for the output of the neural network. Later, the trained neural network generates more data points through an ODE solver without participating in the computational graph, facilitating more accurate estimations of gradients based on Fourier analysis. These two steps form a positive feedback loop, enabling accurate dynamics modeling in our framework. Consequently, our approach outperforms state-of-the-art methods in terms of training time, dynamics prediction, and robustness. Finally, we demonstrate the superior performance of our framework using a number of representative complex systems.

## 1 INTRODUCTION

Complex dynamical systems have garnered considerable attention across various disciplines in the natural and social sciences. These systems are typically characterized by ordinary differential equations (ODEs) or partial differential equations (PDEs) Perko (1991); Meiss (2007). However, the explicit forms of these underlying systems are often completely unknown *a priori*, and the available information about such systems is commonly obtained through experimentally collected time series data. Unfortunately, these data are often noisy and limited due to measurement errors, resource constraints, and the high cost of data collection. As a result, there is an urgent need to uncover the underlying dynamics of these systems without prior knowledge of the specific governing equations, relying solely on the measured time series data Legaard et al. (2021); Zhu et al. (2023). The successful modeling of these systems enables downstream tasks such as prediction Carroll (2018); Pathak et al. (2018) and control Sanhedrai et al. (2020); Zhang & Zhou (2022). Therefore, the development of an efficient and robust data-driven approach for system modeling is of utmost significance.

Recently, there has been a tremendous amount of interest in applying machine learning technologies, such as auto-regressive models Ding et al. (2018), sparsity-promoting methods Kaiser et al. (2017); Meiss (2007), reservoir computing Pathak et al. (2018), and Neural ODEs (NODEs) Chen et al. (2018), to model complex dynamical systems. In particular, NODEs and their extensions Holt et al. (2022); Bilovs et al. (2021); Lanzieri et al. (2022), which involve continuously defined dynamics, have been extensively utilized for coping with the continuous-time datasets due to the direct parameterization of the vector field of ODEs, enabling them to naturally capture the underlying dynamics of continuous-time systems solely based on observational data. Despite their successes, NODEs often encounter challenges related to high computational costs and susceptibility to local optima. During the training process, the NODEs may gradually exhibit complex dynamic behaviors, such as drastic fluctuations or stiffness. These behaviors significantly increase the time required for numerical solving and the memory consumption for backpropagation Chen et al. (2018); Kim et al. (2021); Finlay et al. (2020). Furthermore, the corresponding adjoint method for computing gradients is numerically sensitive Zhuang et al. (2020).

To address the challenges mentioned above and enhance the speed and robustness of dynamic modeling, we propose a framework called Fourier NODEs (FNODEs). The key advantages of our proposed framework are summarized as follows.

- The utilization of Fourier analysis theory provides theoretical guarantees for accurately estimating the gradient flows of dynamical systems with limited and noisy data.

- Directly matching the estimated gradient flows, serving as a simulation-free framework, allows the FNODEs to significantly reduce the training time by more than tenfold compared to the standard NODEs.

- The data augmentation training strategy is introduced by generating more data points for better estimating the gradient flows and then retraining the model, which forms a positive feedback loop, resulting in improved robustness and accuracy in dynamic modeling.

- Our framework can be easily applied for resolution-invariant modeling of PDEs by additionally incorporating the estimated higher-order spatial gradients based on Fourier analysis into the model, expanding the scopes of classical NODEs.

## 2 RELATED WORK

NODEs Chen et al. (2018), a class of continuous-depth neural networks, have garnered substantial interest in recent years due to their pivotal role across various scientific disciplines. The typical residual block, represented as

$$\boldsymbol{z}(t+1) = \boldsymbol{z}(t) + \boldsymbol{F_\theta}[\boldsymbol{z}(t), t],$$

can be construed as an Euler discretization (with a time step $\Delta t = 1$) of an ODE Chen et al. (2018),

$$\dot{\boldsymbol{z}} = \boldsymbol{F_\theta}[\boldsymbol{z}(t), t],$$

where $\boldsymbol{z}(t) \in \mathbb{R}^d$ denotes the state of the $d$-dimensional (-D) system, $t$ is the time, also interpreted as a continuous "depth" in the residual block, and $\boldsymbol{F_\theta} : \mathbb{R}^d \times \mathbb{R} \to \mathbb{R}^d$ can be any neural network architecture with the parameter vector $\boldsymbol{\theta}$ to be optimized. Consequently, given the initial value $\boldsymbol{z}(t_0)$, the state at any time $t_1$ can be computed using an ODE solver:

$$\boldsymbol{z}(t_1) = \text{ODEsolve}[\boldsymbol{z}(t_0), \boldsymbol{F_\theta}, t_0, t_1] = \boldsymbol{z}(t_0) + \int_{t_0}^{t_1} \boldsymbol{F_\theta}[\boldsymbol{z}(t), t] \mathrm{d}t.$$

The successful application of NODEs in modeling continuous time data has led to the emergence of numerous works aimed at addressing the limitations of the original NODEs. These include augmented NODEs Dupont et al. (2019), neural delay differential equations Zhu et al. (2021), neural controlled differential equations Kidger et al. (2020), neural stochastic differential equations Liu et al. (2019), and neural integro-differential equations Zappala et al. (2022). Despite the effectiveness of these methods in handling specific types of dynamical systems, they are not exempt from the challenges of high computational cost during the training process and difficulties in converging to the global optimal solution.

## 3 THE FRAMEWORK OF FNODES

To establish the Fourier NODEs (FNODEs) framework, we consider a controlled dynamical system of the following general form,

$$\dot{\boldsymbol{s}} = \boldsymbol{f}[\boldsymbol{s}(t), \boldsymbol{u}(t)], \tag{1}$$

where $\boldsymbol{s}(t) \in \mathbb{R}^d$ represents the state of the $d$-D system, $\boldsymbol{u}(t) \in \mathbb{R}^m$ is a time-dependent controller, and $\boldsymbol{f} : \mathbb{R}^d \times \mathbb{R}^m \to \mathbb{R}^d$ embodies the inherent dynamics of the system. Typically, the dynamical system is often unknown. Consequently, a common scenario is to describe the underlying dynamics via learning a surrogate model $\boldsymbol{F_\theta}$ parameterized by a neural network solely based on the observational data. In this work, we assume that the controller $\boldsymbol{u}(t)$ is known, and sampled from a specific family of functions.

**The approximation of temporal gradients.** Let $\boldsymbol{h}(t)$ be a continuously differentiable vector-valued function that is $T$-periodic. From a spectral approximation perspective, we can express $\boldsymbol{h}(t)$ as a Fourier series expansion:

$$\boldsymbol{h}(t) = \lim_{K \to \infty} \boldsymbol{H}_K(t) = \lim_{K \to \infty} \left\{ \frac{\boldsymbol{a}_0}{2} + \sum_{k=1}^{K} \left[ \boldsymbol{a}_k \sin\left(\frac{2\pi}{T}kt\right) + \boldsymbol{b}_k \cos\left(\frac{2\pi}{T}kt\right) \right] \right\},$$

where $\boldsymbol{H}_K(t)$ represents the $K$-th partial sum of the Fourier series, $\boldsymbol{a}_k = 2\left[\int_0^T \boldsymbol{h}(t)\cos(kt)\mathrm{d}t\right]/T$, and $\boldsymbol{b}_k = 2\left[\int_0^T \boldsymbol{h}(t)\sin(kt)\mathrm{d}t\right]/T$. We can then utilize the truncated Fourier series to approximate the gradient of $\boldsymbol{h}(t)$. Specifically, we present the Theorem 1, and the proof can be found in Appendix A.1.

**Theorem 1.** *Assume $\boldsymbol{h}(t)$ is a $T$-periodic and thrice differentiable function, and its second derivative $\boldsymbol{h}''(t)$ satisfies the Lipschitz condition on $[0, T]$. Then, for any $\epsilon > 0$, there exists a positive integer $K_0$ such that for any $K > K_0$ and for all $t \in [0, T]$, we have:*

$$\|\boldsymbol{H}_K'(t) - \boldsymbol{h}'(t)\| < \epsilon,$$

*where $\boldsymbol{H}_K'(t) = \frac{2\pi}{T}\sum_{k=1}^{K} k[\boldsymbol{a}_k\sin(2\pi kt/T) + \boldsymbol{b}_k\cos(2\pi kt/T)]$. This implies that $\boldsymbol{H}_K'(t)$ converges uniformly to $\boldsymbol{h}'(t)$ on $[0, T]$.*

In practice, we can collect a discrete-time observational time series $\boldsymbol{S} = \{\boldsymbol{s}_0, \boldsymbol{s}_1, \cdots, \boldsymbol{s}_{N-1}\}$ with a regular sampling rate at times $\{t_0, t_1, \cdots, t_{N-1}\}$. To estimate the Fourier coefficients $\tilde{\boldsymbol{S}} = \{\tilde{\boldsymbol{s}}_0, \tilde{\boldsymbol{s}}_1, \cdots, \tilde{\boldsymbol{s}}_K\}$ from the observational data $\boldsymbol{s}$, we can utilize the discrete Fourier transform (DFT) $\mathscr{F}$ and its inverse DFT (IDFT) $\mathscr{F}^{-1}$ for all $k \in \{0, 1, \cdots, K\}$ and $n \in \{0, 1, \cdots, N-1\}$, given by:

$$\tilde{\boldsymbol{s}}_k = \mathscr{F}(\boldsymbol{S}, k) = \sum_{n=0}^{N-1} \boldsymbol{s}_n \mathrm{e}^{-i\frac{2\pi n}{N}k}, \quad \boldsymbol{s}_n = \mathscr{F}^{-1}(\tilde{\boldsymbol{S}}, n) = \frac{1}{N}\sum_{k=0}^{K} \tilde{\boldsymbol{s}}_k \mathrm{e}^{i\frac{2\pi k}{N}n}. \tag{2}$$

According to the Nyquist sampling theorem, when $N > 2K$, we can estimate the temporal derivative of the state at time $t_n$, which can be expressed as:

$$\hat{\boldsymbol{s}}'(t_n) = \frac{1}{N}\sum_{k=0}^{K} i\frac{2\pi k}{T}\tilde{\boldsymbol{s}}_k \mathrm{e}^{i\frac{2\pi k}{N}n} \approx \boldsymbol{s}'(t_n), \quad n \in \{0, 1, \cdots, N-1\}. \tag{3}$$

**The approximation of spatial gradients of PDEs.** When considering systems described by PDEs, the state of the system evolves over both time and space, with the underlying dynamics often influenced by spatial gradients. In such cases, the ODE in Eq. (1) can be extended to:

$$\partial_t \boldsymbol{s} = \boldsymbol{f}(\boldsymbol{s}, \boldsymbol{u}, \partial_x \boldsymbol{s}, \partial_y \boldsymbol{s}, \partial_{xx} \boldsymbol{s}, \cdots), \quad \boldsymbol{x} = \{x, y, \cdots\} \in \Omega,$$
$$\boldsymbol{s}(\boldsymbol{x}, 0) = s_0, \quad \boldsymbol{x} \in \Omega,$$

where $\Omega$ represents the spatial domain $\boldsymbol{x}$, and the system exhibits periodic boundary conditions. To accurately model the underlying dynamics $\boldsymbol{f}$, it is necessary to estimate the spatial gradients in advance. Let's consider a PDE with two spatial dimensions, denoted by $x$ and $y$. The system's state at time $t$ is sampled at the values $\boldsymbol{s}_{j_x j_y}(t) = \boldsymbol{s}(x_{j_x}, y_{j_y}, t)$, where $j_x \in \{0, 1, \ldots, N_x - 1\}$ and $j_y \in \{0, 1, \ldots, N_y - 1\}$. This leads to the following expressions:

$$\tilde{\boldsymbol{s}}_{k_x k_y} = \sum_{j_x=0}^{N_x-1}\sum_{j_y=0}^{N_y-1} \boldsymbol{s}_{j_x j_y} \mathrm{e}^{-i2\pi\left(\frac{j_x k_x}{N_x} + \frac{j_y k_y}{N_y}\right)}, \quad \boldsymbol{s}_{j_x j_y} = \frac{1}{N_x N_y}\sum_{k_x=0}^{K_x}\sum_{k_y=0}^{K_y} \tilde{\boldsymbol{s}}_{k_x k_y} \mathrm{e}^{i2\pi\left(\frac{k_x j_x}{N_x} + \frac{k_y j_y}{N_y}\right)},$$

where $K_x$ and $K_y$ represent the cutoff frequencies in the dimensions of $x$ and $y$, respectively.

Denote $L_x$ and $L_y$ as the periodic lengths in the dimensions of $x$ and $y$, respectively. And let $\boldsymbol{e}_M = (0, 1, \cdots, M-1)^\top$, $\boldsymbol{k_x} = \frac{2\pi}{L_x}(\boldsymbol{e}_{L_x}, \cdots, \boldsymbol{e}_{L_x})$, and $\boldsymbol{k_y} = \frac{2\pi}{L_y}(\boldsymbol{e}_{L_y}, \cdots, \boldsymbol{e}_{L_y})^\top$. Similar to estimating temporal gradients, we can estimate the spatial gradients for all $p, q \in \{0, 1, \cdots\}$ using the following equation:

$$\frac{\partial^{p+q}\boldsymbol{S}_t}{\partial^p x \partial^q y} = \mathscr{F}_{xy}^{-1}\left[(i\boldsymbol{k_x})^p (i\boldsymbol{k_y})^q \mathscr{F}_{xy}(\boldsymbol{S}_t)\right], \tag{4}$$

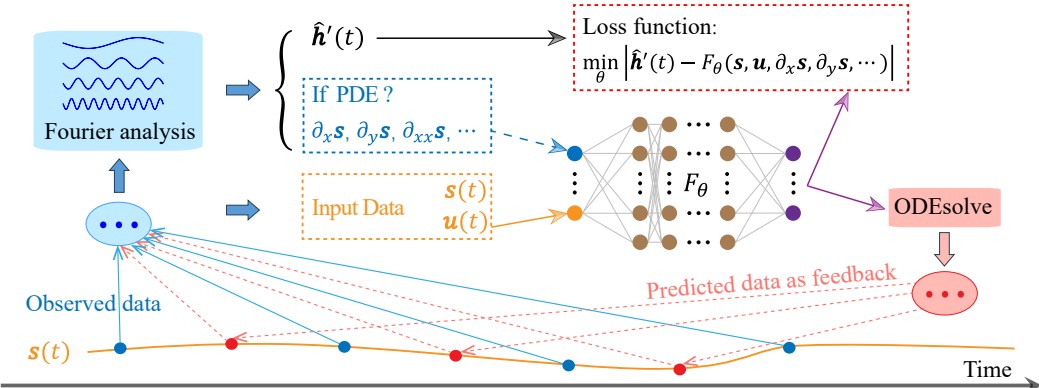

Figure 1: Illustration of the FNODEs framework.

where $\boldsymbol{S}_t$ is the sampling matrix of the system at time $t$, with elements $\boldsymbol{S}_t[j_x, j_y] = \boldsymbol{s}_{j_x j_y}(t)$. $\mathscr{F}_{xy}$ and $\mathscr{F}_{xy}^{-1}$ represent the 2-D DFT and IDFT in space, respectively.

**The framework of FNODEs.** Unlike classical NODEs, we introduce the FNODEs framework, which utilizes spatiotemporal gradients obtained from Fourier analysis for more precise dynamical learning. As depicted in Figure 1, we start by estimating the gradients in the temporal and spatial directions using Eqs. (3) and (4) based on the observational time series $\boldsymbol{S}$ and the corresponding control signal $\boldsymbol{U} = \{\boldsymbol{u}_0, \boldsymbol{u}_1, \cdots, \boldsymbol{u}_{N-1}\}$. Next, we construct a neural network $\boldsymbol{F}_{\boldsymbol{\theta}}$ to learn the underlying dynamics of the system. Specifically, for any time $t$, this neural network takes the system state $\boldsymbol{s}$, control signal $\boldsymbol{u}$, and spatial gradients $\{\partial_x \boldsymbol{s}, \partial_y \boldsymbol{s}, \cdots\}$ (only for PDEs) as inputs, and outputs the prediction of the temporal gradient at time $t$. During this process, we update the parameters $\boldsymbol{\theta}$ using the following gradient flow matching loss function:

$$\mathcal{L} = \frac{1}{N} \sum_{n=0}^{N-1} \left\| \hat{\boldsymbol{h}}'(t_n) - \boldsymbol{F}_{\boldsymbol{\theta}}[\boldsymbol{s}(t_n), \boldsymbol{u}(t_n), \partial_x \boldsymbol{s}(t_n), \partial_y \boldsymbol{s}(t_n), \cdots] \right\|. \tag{5}$$

Notably, in the training process, there is no need to utilize the ODE solvers, resulting in a simulation-free training framework.

**Data augmentation.** In addition, we incorporate a data augmentation strategy to facilitate the modeling of complex systems. This strategy involves simulating the trained NODEs to obtain the system's states at arbitrary times, thereby increasing the sampling frequency. By generating a larger sample size, we can better satisfy the requirements of the Nyquist sampling theorem, leading to more accurate estimations of the spatiotemporal gradients. The newly generated data is then fed into our model for further training. This two-step process, consisting of data augmentation and training, forms a positive feedback loop that enhances the accuracy and robustness of our approach.

It is important to note that the augmentation strategy does not participate in the computational graph, resulting in a low computational cost limited to the forward simulation. In contrast, the classical NODEs framework includes simulation in the training process, which incurs a high computational cost due to solving extremely high-dimensional ODEs in the backward pass. The entire framework is illustrated in Fig. 1, and we provide a detailed execution process in Algorithm 1 in Appendix A.2.

## 4 ILLUSTRATIVE EXPERIMENTS

In this section, we present a thorough analysis of our framework based on experiments conducted using a computational setup with 64GB RAM and an NVIDIA Tesla V100 GPU equipped with 16GB memory. Specifically, we generate a set of control functions $u$ using Gaussian random fields (GRFs) with a radial-basis function kernel, as defined by

$$u \sim c \cdot \mathcal{G}\left\{\mu, \exp\left[\|x_1 - x_2\|^2 / (2l^2)\right]\right\},$$

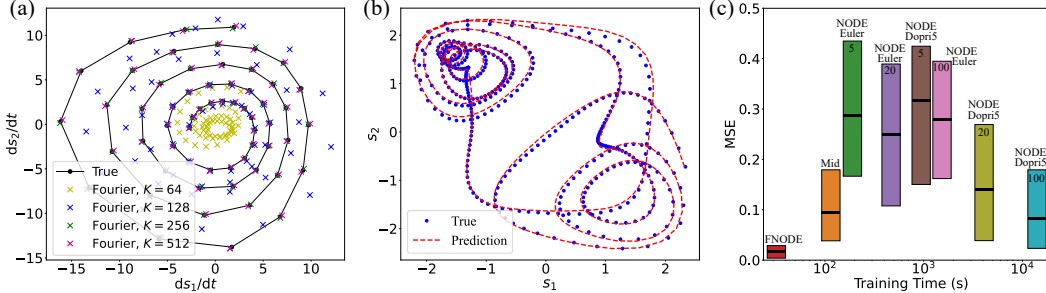

Figure 2: The experimental results of system (6) using FNODEs. (a) Estimation of the derivatives based on the observational data using Fourier analysis. (b) The test prediction using the trained model. (c) Comparison of the FNODEs and baseline methods in terms of training time and prediction error.

where $\mu$ represents the mean value, $l$ is the length scale that governs the smoothness of the sampling function, and $c$ is the scaling factor of the output. To validate our framework, we consider experimental data $s(\boldsymbol{x}, t, \boldsymbol{u}(\boldsymbol{x}, t))$ derived from multiple dynamical systems, and compare our results with several state-of-the-art baseline methods, including NODEs Chen et al. (2018), Deep Operator Network (DeepONet) Lu et al. (2019), Fourier Neural Operator (FNO) Li et al. (2020), PDE-Net Long et al. (2018), Physics-Informed Neural Networks (PINN) Raissi et al. (2019), Deep Auto-Regressive (DeepAR) Flunkert et al. (2017), and the message-passing PDE solvers (MP-PDE) Brandstetter et al. (2022). Detailed descriptions of these baseline methods can be found in Appendix D.

## 4.1 PARAMETRIC ODES

We initially consider a 2-D parametric ODE system, which is described by

$$\frac{\mathrm{d}\boldsymbol{s}}{\mathrm{d}t} = \begin{bmatrix} -0.1 & 2.0 \\ -2.0 & -0.1 \end{bmatrix} \boldsymbol{s}^3 + \boldsymbol{u}(t), \quad \boldsymbol{s}(0) = \boldsymbol{s}_0 \tag{6}$$

where $\boldsymbol{u}(t) = [u_1(t), u_2(t)]^\top$ is a vector-valued function derived from Gaussian Random Fields (GRF). We generate a dataset comprising 100 training samples, 30 validation samples, and 50 testing samples, with $\mu = 0$, $c = 100$, $l = 0.1$, and random initial values $\boldsymbol{s}_0$.

Here, we randomly select a segment of data $\boldsymbol{S}_g$, and each temporal gradient of the state at the time $t$ in $\boldsymbol{S}_g$ is estimated directly using Eq. (3). The estimation of temporal gradients for different truncation frequencies $K$ is illustrated in Figure 2(a). It is evident that Eq. (3) is not effective in approximating the true gradients when $K$ is small. However, when $K$ exceeds a certain threshold, Fourier analysis can effectively approximate the true gradients. In our experiment, with $N = 500$ sampling points, the maximum value of $K$ for recoverable signals, as per the Nyquist sampling theorem, is 250. Therefore, setting $K$ to 512 does not yield better performance compared to setting it to 256. Moreover, our framework exhibits robustness to approximation errors in gradient estimation from a statistical perspective due to the inclusion of multiple data points under different control signals in the training set.

Table 1: Comparison of the prediction MSE and the training time of baseline methods under different training set sizes and noise intensities. Here, we consider zero-mean Gaussian noise with $N_{\mathrm{tr}} = 1000$, and the standard deviation is $\sigma_{\mathrm{sd}}$ times the mean absolute value of the data.

| Models | Training set sizes $N_{\mathrm{tr}}$ | | | | Noise intensify $\sigma_{\mathrm{sd}}$ | | | | Training time (s) |
|---|---|---|---|---|---|---|---|---|---|
| | 10 | 20 | 100 | 1000 | 0.1% | 1% | 5% | 10% | |
| Mid | 0.129 | 0.102 | 0.075 | 0.072 | 0.103 | 0.115 | 0.134 | 0.285 | 320.1 |
| DeepAR | 3.467 | 2.318 | 0.633 | 0.801 | 0.783 | 0.884 | 1.386 | 2.297 | 101.8 |
| NODE (Euler) | 0.211 | 0.191 | 0.157 | 0.132 | 0.133 | 0.193 | 0.220 | 0.280 | 631.8 |
| NODE (Dopri5) | 0.156 | 0.096 | 0.087 | 0.062 | 0.111 | 0.172 | 0.188 | 0.264 | 9632.2 |
| FNODE | **0.049** | **0.030** | **0.018** | **0.012** | **0.016** | **0.013** | **0.083** | **0.184** | **62.4** |

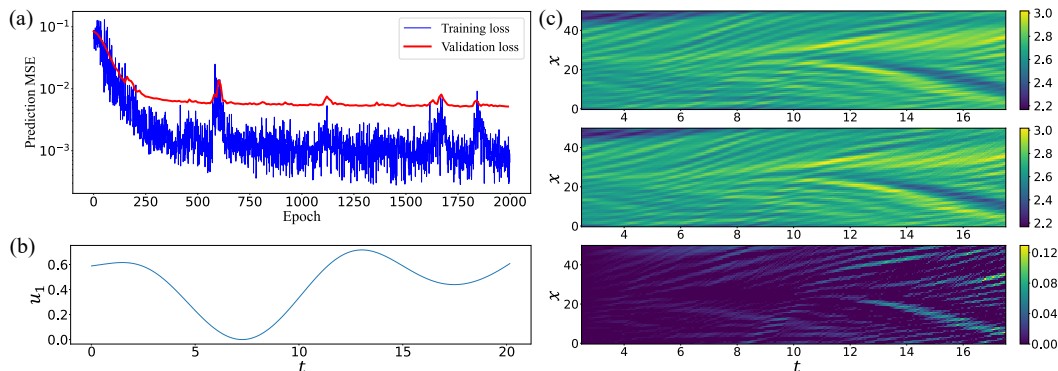

Figure 3: The experimental results of the KDV system using FNODEs Method. (a) The variation of the training and validation loss during the training process. (b) Plot of $u_1(t)$ from a test data. (c) These three subplots from top to bottom represent the ground truth, predicted results, and prediction errors of test data.

After training our model, we assess its performance on the test data. As depicted in Figure 2(b), our framework consistently achieves high prediction accuracy for long-term forecasting. To further demonstrate its effectiveness, we introduce a new baseline method called the "Mid" method, which estimates gradients using the classical central difference numerical method. Additionally, we conduct experiments using the NODEs with two numerical ODE solvers, namely "Euler" and "Dopri". Figure 2(c) displays the training time and the prediction mean squared error (MSE) of the different baselines on the test data. It is evident that the FNODEs require the shortest training time and simultaneously achieve the lowest prediction error.

We note that the classical NODE method tends to get trapped in local optima in this particular example, resulting in longer training times. To further verify the robustness of our proposed framework, we conduct experiments with varying training set sizes and noise intensities. Table 1 demonstrates that the FNODEs outperform the baselines in our experiments, showcasing stronger robustness and superior prediction capabilities.

## 4.2 PARAMETRIC PDEs

To further assess the effectiveness of our framework, we conduct experiments on various common parametric PDEs, including the Korteweg-de Vries (KDV) system, diffusion-reaction (DR) system, Kuramoto-Sivashinsky (KS) system, and a 2-D Navier-Stokes (NS) system. These systems are summarized in Table 2.

Table 2: The explicit forms of four parametric PDEs considered in our experiments.

| Systems | Equations | Domain |
|---------|-----------|--------|
| KDV | $\partial_t s = \partial_{xxx} s + u(x,t)\partial_x(s^2/2)$ | $x \in [0, 16\pi],\ t \in [0, 20]$ |
| DR | $\partial_t s = 0.01\partial_{xx} s + 0.01s^2 + u(x,t)$ | $x \in [0, 1],\ t \in [0, 1]$ |
| KS | $\partial_t s = -\partial_x(s^2/2) - \partial_{xx} s - u(x,t)\partial_{xxxx} s$ | $x \in [0, 32\pi],\ t \in [0, 20]$ |
| NS | $\partial_t s = \partial_x \gamma \partial_y s - \partial_y \gamma \partial_x s + 0.001\Delta s + u(x,y,t),\ \Delta\gamma = -s$ | $(x,y) \in [0, 2]^2,\ t \in [0, 10]$ |

**Korteweg-de Vries system.** We begin by considering the KDV system, which is commonly used to describe the evolution of water waves. In this equation, $s(x, t)$ represents the wave displacement, $x$ denotes the spatial coordinate and $t$ represents time. The control parameter is denoted as $u(x, t)$. In this example, we consider $u(x, t) = \sin(x/8) + u_1(t)$, where $u_1(t)$ is a 1-D function sampled from GRFs. To validate the effectiveness of our framework, we fix the initial value $s_0$ and generate 100 training data, 30 validation data, and 30 test data. During the training process, we estimate the spatial partial derivatives $\mathscr{F}^{-1}\{i\boldsymbol{k_x}, (i\boldsymbol{k_x})^2, (i\boldsymbol{k_x})^3, (i\boldsymbol{k_x})^4\}$ using Eq. (4) as additional inputs to the neural network $\boldsymbol{F_\theta}$. Then, we employ the gradient flow matching loss Eq. (5) to train our model.

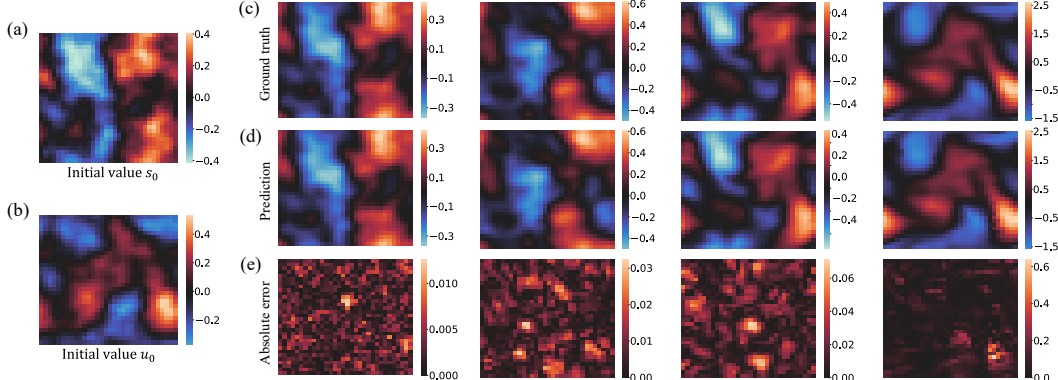

Figure 4: The experimental results of the NS system using FNODEs Method. (a) The initial state $s_0$ of a certain test data. (b) The initial state $u_0$ of the parameter $u(x, y, 0)$. (c)-(e) These subfigures depict the ground truth, prediction results, and prediction errors at different time instances. From left to right, the time instances $t$ correspond to 2.5s, 5s, 7.5s, and 10s.

During the training process, both the training loss and validation loss rapidly decrease to lower levels (see Fig. 3(a)). The entire training process is completed in just 28 seconds for 2000 epochs, indicating the efficiency of our framework. In addition, our framework exhibits outstanding predictive performance on the test data with an average prediction error of only 0.04. Figures 3(b)-(c) depict the prediction results for one sample of the test data set, clearly demonstrating the sustained high prediction accuracy over an extended period, successfully revealing the underlying dynamics of the KDV system. For the 1-D DR and KS, we provide the corresponding results in Appendix C in a similar manner.

**Navier-Stokes system.** Next, we turn our attention to the more complex 2-D NS system in the vorticity form, which involves two spatial dimensions, namely $x$ and $y$. In this case, we consider $u(x, y, t) = u_0(x, y) + u_1(t)$, where $u_0$ and $u_1$ are 1-D functions sampled from GRFs. Similar to the process in the KDV system, we incorporate the estimated spatial partial derivatives as inputs into the neural network using Eq. (4).

In addition to the conventional terms $\mathscr{F}^{-1}\{(i\boldsymbol{k_x}), (i\boldsymbol{k_y}), (i\boldsymbol{k_x})^2, (i\boldsymbol{k_y})^2, (-\boldsymbol{k_x}\boldsymbol{k_y}), \cdots\}$, we introduce two additional prior terms $\mathscr{F}^{-1}[i\boldsymbol{k_x}/(\boldsymbol{k_x^2} + \boldsymbol{k_y^2})]$ and $\mathscr{F}^{-1}[i\boldsymbol{k_y}/(\boldsymbol{k_x^2} + \boldsymbol{k_y^2})]$ to better capture the intrinsic variable $\gamma$ in the equation. Consequently, our framework successfully learns the dynamics of the NS system, achieving an average prediction error of only 0.03. We present the predicted results of the test data in Fig. 4, which clearly demonstrate the excellent predictive performance of our framework within the specified time range. Moreover, the FNODEs exhibit fast training speed in a runtime of 283 seconds for 5000 epochs.

**Evaluation against baseline methods.** To substantiate the efficacy of our proposed framework, we compare it with the state-of-the-art techniques, considering factors such as the size of the training set, robustness to noise, and training time. Specifically, we conduct experiments on four distinct PDE systems under three varying experimental configurations, and the results are shown in Table 3. It is unequivocally clear that our framework is capable of discerning the underlying dynamics in the shortest time, particularly surpassing the standard NODEs method by more than 10 times. For an in-depth understanding of the baseline methods, please refer to Appendix D.

Table 3: Comparing the training time of different methods in multiple experiments.

| Systems | NODEs | DeepONet | FNO | PDE-NET | PINN | MP-PDE | FNODEs |
|---------|-------|----------|-----|---------|------|--------|--------|
| KDV | 3938 | 751 | 271 | 398 | 1540 | 560 | **130** |
| DR | 2910 | 652 | 262 | 311 | 1532 | 367 | **65** |
| KS | 5120 | 641 | 241 | 492 | 1660 | 589 | **74** |
| NS | 21747 | 8484 | 549 | 1323 | 2430 | 734 | **283** |

Furthermore, Table 4 illustrates the MSE of the test data under diverse experimental configurations. It is observed that our proposed framework consistently yields lower prediction errors in comparison to the baseline methods. Here, it should be noted that our framework may require a longer prediction time compared to neural operator methods such as DeepONet and FNO, it exhibits superior performance in terms of the generalization, especially for NS systems with random initial values. Additionally, our framework surpasses the performance of the PDE-NET, thereby further validating the effectiveness of Fourier analysis-based estimation of spatiotemporal gradients over traditional numerical gradient estimation methods. As for the MP-PDE method, as a black-box method in deep learning, it did not demonstrate satisfactory predictive performance in our experiments with a limited training set. In addition, the PINN method demonstrates performance similar to our approach, exhibiting robustness against fluctuations in training set size and noise levels. This is attributable to the necessity of employing underlying dynamical equations for training. Therefore, in data-driven modelling tasks (PINN is not applicable), our method indeed learned the underlying dynamics, demonstrating significant advantages in predictive capability.

Table 4: Prediction MSE under different methods, systems, and experimental conditions.

| Models | $N_{tr} = 10$, $\sigma_{sd} = 0$ | | | | $N_{tr} = 1000$, $\sigma_{sd} = 0$ | | | | $N_{tr} = 1000$, $\sigma_{sd} = 5\%$ | | | |
|---|---|---|---|---|---|---|---|---|---|---|---|---|
| | KDV | DR | KS | NS | KDV | DR | KS | NS | KDV | DR | KS | NS |
| NODEs | 1.93 | 1.68 | 2.56 | 4.21 | 0.45 | 0.58 | 1.69 | 3.61 | 1.02 | 1.46 | 2.34 | 5.19 |
| DeepONet | 0.83 | 2.75 | 3.02 | 38.3 | 0.09 | 0.17 | 0.63 | 7.30 | 0.74 | 0.87 | 1.20 | 14.7 |
| FNO | 1.88 | 1.56 | 2.21 | 23.9 | 0.21 | 0.13 | 0.92 | 5.3 | 0.89 | 1.58 | 2.13 | 12.6 |
| PDE-NET | 1.64 | 1.38 | 2.32 | 2.09 | 0.11 | 0.27 | 0.78 | 0.92 | 1.34 | 0.97 | 1.73 | 2.30 |
| PINN | **0.17** | **0.19** | **0.24** | **0.31** | **0.17** | **0.19** | **0.24** | **0.31** | **0.17** | **0.19** | **0.24** | **0.31** |
| MP-PDE | 2.39 | 2.74 | 4.23 | 6.62 | 0.44 | 0.49 | 1.33 | 2.07 | 1.22 | 1.11 | 1.93 | 2.59 |
| FNODEs | **0.23** | **0.16** | **0.28** | **0.29** | **0.023** | **0.016** | **0.11** | **0.083** | **0.14** | **0.12** | **0.46** | **0.11** |

### 4.3 ADDITIONAL VALIDATION AND DISCUSSION OF OUR FRAMEWORK

**Handling Sparse and Irregularly Sampled Time Series.** Our framework can be extended to the case of irregularly sampled time series by replacing Eqs. (2) and (3) with the nonuniform discrete Fourier transform (NDFT). The computation of NDFT can be expedited using the nonuniform fast Fourier transform (NFFT) Dutt & Rokhlin (1993); Barnett et al. (2018). Moreover, in scenarios where the number of sampling data points is limited, the restorable signal frequency $N/2$, as per the Nyquist sampling theorem, may be less than the cutoff frequency $K$ of the true signal. Under such circumstances, Fourier analysis may fail to accurately estimate the temporal gradients of the system. However, our data augmentation training strategy can circumvent this obstacle by integrating the predicted results of the FNODEs into the original dataset, thereby significantly enhancing the data utilization.

**Resolution-invariant property of the FNODEs.**

To investigate the relationship between the truncated frequency $K$ and the number of sampling points $N$, we conduct experiments on the system (6) by generating data with varying sparsity levels, while maintaining other hyperparameters consistent with the previous section. As depicted in Fig. 5(a), we employ Fourier analysis to estimate the temporal gradients at different sparsity levels. The optimal truncated frequency for each dataset is almost near $N/2$, which underscores the non-negligible role of Fourier basis with $K < 256$ in system modeling. Therefore, a larger number of sampling points $N$ allows for a higher usable cutoff frequency $K$, facilitating the capture of the true signal gradients.

Then, we initially consider 100 training time series with only 200 sampling points with $K = 100$. Under this setting, the prediction of the trained model on test data is displayed in Fig. 5(c). Subsequently, we use the trained model to predict 200 additional data points to augment the training data and retrain our model with $K = 200$. The prediction result in this case is depicted in Fig. 5(d). In the same manner, we further predict 400 points and set $K = 400$ to train our model. The corresponding predicted result is shown in Fig. 5(e). Moreover, Figure 5(b) presents the boxplots of prediction errors on all test data under previous settings. It is clear that the data augmentation training strategy

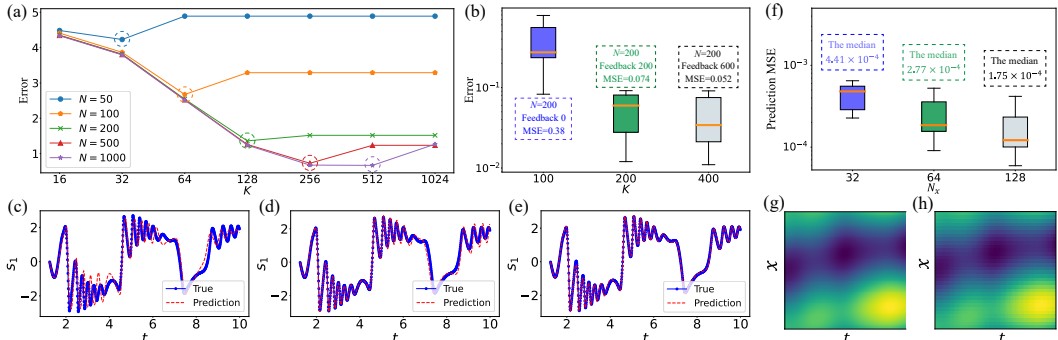

Figure 5: Experimental results of the data augmentation and the resolution-invariant predictions using the FNODEs. In system (6), (a) Prediction errors of time gradients under different sampling rates $N$ and cutoff frequencies $K$. (b) A boxplot depicting the testing errors at different stages under the utilization of feedback data, with the median of prediction errors represented by a yellow horizontal line. (c)-(e) The prediction results of test data under different feedback settings (0, 200, and 600 feedback points respectively). In DR system, (f) The prediction MSE of temporal gradient under different $N_x$ in training data. (g) A training sample with a low resolution. (h) A predicted sample with a high resolution.

can further enhance the modeling performance without measuring additional data samples, which may be prohibited in the real-world experiments.

Indeed, our framework can be directly employed for resolution-invariant modeling of PDE systems. In the temporal dimension, the employment of ODEsolve in the trained model can predict the system state at an arbitrary time point. For the spatial dimension, our model can be trained on data with lower sampling frequencies but make predictions on data with higher sampling frequencies. To verify this property, we train our model on training data of the DR system with $N_x$ equal to 32, 64, and 128, respectively, and make predictions on the test set with $N_x = 128$. The corresponding prediction errors are shown in Fig. 5(f). Figure 5(g), as a case study, is the prediction with the number of spatial samples $N_x = 128$ based on the trained model using the lower sampling frequency data with $N_x = 32$ as shown in Fig. 5(h). It is evident that our framework can achieve effective predictions of PDEs with zero-shot super-resolution.

## 5 CONCLUDING REMARKS

In this article, we propose the FNODEs framework, a novel approach for efficient and robust dynamical system modeling. Our framework addresses the challenges faced by standard NODEs, including high computational cost and susceptibility to local optima. The proposed framework combines Fourier analysis and NODEs to transform the modeling problem of complex dynamics into a gradient flow matching task.

We validate our framework on multiple ODE and PDE systems, and experimental results demonstrate that our framework exhibits significantly faster training speed, surpassing the standard NODE by more than tenfold. The FNODEs outperform NODEs and other state-of-the-art methods in various prediction tasks, particularly demonstrating strong robustness in noisy data. Furthermore, we validate, both theoretically and experimentally, that the data augmentation training strategy extremely enhances the maximum truncated frequency of Fourier analysis, facilitating the spatiotemporal gradient estimations and system modeling.

However, our method has certain limitations. For instance, estimating the gradient flow within an acceptable truncated frequency using Fourier analysis becomes challenging when dealing with more complex systems. One potential direction for resolution is to extend our framework by combining Koopman and uncertainty quantification theories. Additionally, our method exhibits inefficiency similar to NODEs when predicting high-dimensional systems, necessitating further optimization by integrating the idea of neural operators.

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

**Appendix**

## A  THEOREMS AND ALGORITHMS

### A.1  PROOF OF THEOREM 1

We aim to prove that the derivative of the truncated Fourier series converges uniformly to $\boldsymbol{h}'(t)$. Let's denote $\boldsymbol{H}_K(t)$ as the partial sum of the Fourier series of $\boldsymbol{h}(t)$:

$$\boldsymbol{H}_K(t) = \boldsymbol{a}_0 + \sum_{k=1}^{K}[\boldsymbol{a}_k \cos(\frac{2\pi}{T}kt) + \boldsymbol{b}_k \sin(\frac{2\pi}{T}kt)], \ \ \boldsymbol{h}'(t) = \frac{2\pi}{T}\sum_{k=1}^{\infty}[k\boldsymbol{b}_k \cos(\frac{2\pi}{T}kt) - k\boldsymbol{a}_k \sin(\frac{2\pi}{T}kt)].$$

We consider the derivative of the partial sum of the Fourier series of $\boldsymbol{h}(t)$, denoted as $\boldsymbol{H}'_K(t)$:

$$\boldsymbol{H}'_K(t) = \frac{2\pi}{T}\sum_{k=1}^{K}[k\boldsymbol{b}_k \cos(\frac{2\pi}{T}kt) - k\boldsymbol{a}_k \sin(\frac{2\pi}{T}kt)].$$

We define $\boldsymbol{G}_K(t) = \boldsymbol{h}'(t) - \boldsymbol{H}'_K(t)$, so we have:

$$\boldsymbol{G}_K(t) = \frac{2\pi}{T}\sum_{k=K+1}^{\infty}[k\boldsymbol{b}_k \cos(\frac{2\pi}{T}kt) - k\boldsymbol{a}_k \sin(\frac{2\pi}{T}kt)].$$

Since $\boldsymbol{h}''(t)$ satisfies the Lipschitz condition, there exists a Lipschitz constant $M$ such that:

$$\|\boldsymbol{h}''(t_1) - \boldsymbol{h}''(t_2)\| \le M_0|t_1 - t_2|, \quad \forall t_1, t_2 \in [-\pi, \pi],$$

where $M_0$ is a positive real number. Given the existence of $\boldsymbol{h}$'s third derivative, we have:

$$|\boldsymbol{h}'''(t)| = \lim_{h \to 0}\left\|\frac{\boldsymbol{h}''(t+h) - \boldsymbol{h}''(t)}{h}\right\| \le M, \quad t \in [-\pi, \pi).$$

Consequently, $M_0$ also serves as the upper bound of the third-order derivative of $\boldsymbol{h}$. Now, we can find the upper bounds for $|\boldsymbol{a}_k|$ and $|\boldsymbol{b}_k|$. We have:

$$\boldsymbol{a}_k = \frac{2}{T}\int_0^T \boldsymbol{h}(t)\cos(\frac{2\pi}{T}kt)dt, \quad \boldsymbol{b}_k = \frac{2}{T}\int_0^T \boldsymbol{h}(t)\sin(\frac{2\pi}{T}kt)dt.$$

Integrating by parts three times, we get:

$$\boldsymbol{a}_k = \frac{1}{k^3} \times \frac{T^2}{4\pi^3}\int_{-\pi}^{\pi} \boldsymbol{h}'''(t)\cos(\frac{2\pi}{T}kt)dt, \quad \boldsymbol{b}_k = -\frac{1}{k^3} \times \frac{T^2}{4\pi^3}\int_{-\pi}^{\pi} \boldsymbol{h}'''(t)\sin(\frac{2\pi}{T}kt)dt.$$

Using the triangle inequality, we have:

$$|\boldsymbol{a}_k| \le \frac{M_1}{k^3}\int_0^T |\boldsymbol{h}'''(t)|dt, \quad |\boldsymbol{b}_k| \le \frac{M_1}{k^3}\int_0^T |\boldsymbol{h}'''(t)|dt,$$

where $M_1 = T^2/(4\pi^3) > 0$. Then we have:

$$|\boldsymbol{a}_k|, |\boldsymbol{b}_k| \le \frac{M_0 M_1 T}{k^3}.$$

Let $M = M_0 M_1 T > 0$, we get:

$$|\boldsymbol{G}_N(t)| \le \sum_{k=K+1}^{\infty}\frac{M}{k^3}|k\cos(kt) - k\sin(kt)| \le \sum_{k=K+1}^{\infty}\frac{M}{k^2}.$$

The right-hand side is a $p$-series, where $p = 2 > 1$, which is a convergent series. Therefore, as $K \to \infty$, we have $|\boldsymbol{G}_K(t)| \to 0$. This means that $\boldsymbol{H}'_K(t)$ converges uniformly to $\boldsymbol{h}'(t)$. This completes the proof.

---

**Algorithm 1:** Execution process of the FNODEs framework

---

**Data:** The observed data $s(t_n)$, $n \in \{0, 1, \cdots, N-1\}$; Control parameter $u(t)$.

**Result:** The trained agent model $F_\theta$.

**Step1:** Set an appropriate truncation frequency $K$, neural network $F_\theta$ ;

**Step2:** Estimate the temporal gradient $\hat{h}'(t_n)$ using Eqs. (2) and (3) of the main text, and the spatial gradient (if it is a PDE) using Eqs. (5) and (6) of the main text;

**Step3:** Compute the loss function using Eq. (7) of the main text and update the parameters $\theta$ ;

**Step4:** If the prediction results of $F_\theta$ have a validation loss below a certain threshold, the algorithm terminates; otherwise, proceed to the next step ;

**Step5:** Predict $s((t_n + t_{n+1})/2)$ using $F_\theta$, where $n \in \{0, 1, \cdots, N-2\}$ ;

**Step6:** Let $N \leftarrow 2N - 1$ and merge the observed data and predicted data in chronological order, still denoted as $s(t_n)$, $n \in \{0, 1, \cdots, N-1\}$. Return to Step 2.

---

Table S1: Experimental hyperparameters in different systems

| Experiment | $N_{\text{tr}}$ | $N$ | $N_{\boldsymbol{x}}$ | $d_i$ | $\mu$ | $l$ | $c$ | $K$ | $l_h$ |
|---|---|---|---|---|---|---|---|---|---|
| 2-D ODE | 100 | 1000 | / | 3 | 0 | 0.1 | 20 | 300 | 3 |
| KDV PDE | 100 | 1000 | 64 | 6 | 0 | 0.2 | 1.0 | 512 | 3 |
| DR PDE | 100 | 1000 | 64 | 6 | 0 | 0.1 | 1.0 | 512 | 3 |
| KS PDE | 100 | 1000 | 64 | 6 | 2 | 0.2 | 1.0 | 512 | 3 |
| NS PDE | 100 | 1000 | $32 \times 32$ | 8 | 0 | 0.1 | 1.0 | 256 | 4 |

A.2 EXECUTION PROCESS OF THE FNODES FRAMEWORK

To provide a clearer description of our method's execution process, we present a concise procedure as Algorithm 1.

# B EXPERIMENTAL HYPERPARAMETERS

In this section, we present the hyperparameter settings for all experiments conducted in the main text and appendix. First, we construct a multi-layer fully connected neural network to model the underlying dynamics $\boldsymbol{f}$, and use the neural network to predict the vector field of the dynamical system from a dynamical perspective. The neural network consists of four hyperparameters: the input dimension $d_i$, the hidden layer dimension $d_h$, the number of hidden layers $l_h$, and the output dimension $d_o$. Regarding the training data, it comprises three hyperparameters: the number of training samples $N_{\text{tr}}$, the number of time domain sampling points $N$, and the number of spatial domain sampling points $N_x$ and/or $N_y$. For Fourier analysis, it is necessary to consider the truncation frequency $K$. For the training process, In Gaussian random field sampling, we consider the output scale $c$, mean $\mu$, and length scale $l$. Therefore, unless otherwise specified, the hyperparameter settings for our experiments are shown in Table S1.

During the training process, we set the learning rate to 0.001, weight decay to 1e-5, and employed the Adam optimizer for training. In addition, the initial values for the NS and 2D-ODE experiments are randomly generated, while for the KDV, DR, and KS experiments, the initial values are given as follows:

$$s_{\text{KDV}} = 2\cos(x/L_{\text{KDV}}) \times (1 + \sin(x/L_{\text{KDV}})),$$
$$s_{\text{DR}} = \cos(2\pi x/L_{\text{DR}}),$$
$$s_{\text{KS}} = 2\cos(x/L_{\text{KS}}) \times (1 + \sin(x/L_{\text{KS}})),$$

where $L_{\text{KDV}}$, $L_{\text{DR}}$, and $L_{\text{KS}}$ represent the spatial lengths of the KDV, DR, and KS systems, respectively.

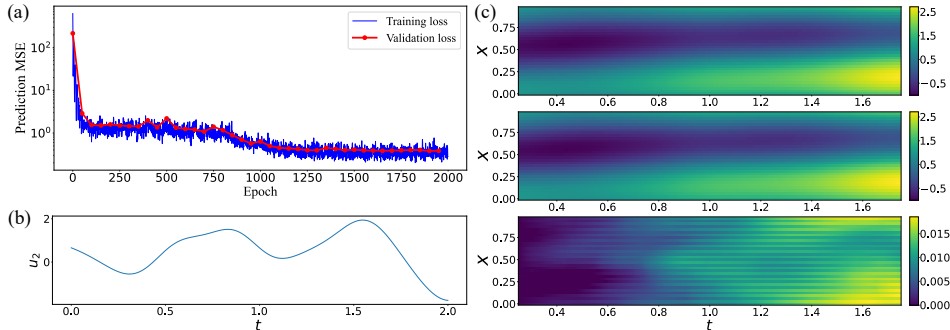

Figure S1: The experimental results of the DR system using FNODEs Method. (a) The variation of the training and validation loss during the training process. (b) Plot of $u_2(t)$ from a test data. (c) These three subplots from top to bottom represent the ground truth, predicted results, and prediction errors of test data.

## C    EXPERIMENTAL DETAILS

In this section, we provide additional experiments and explanations to illustrate the content in the main text better.

### C.1    POTENTIAL SPATIAL GRADIENTS

To improve the modeling of PDE systems, we consider the potential spatial derivative terms estimated from equations (7) and (8) as additional inputs to the neural network. In fact, these partial derivatives are commonly used in the majority of PDE systems. For instance, in one-dimensional PDE systems, the most prevalent spatial partial derivatives are $\{\partial_x s, \partial_{xx} s, \partial_{xxx} s, \ldots\}$, and the highest order $d$ of the partial derivatives is typically chosen to be a small finite value (we set it to 4 in our experiments in this paper). Therefore, for a PDE system with a spatial dimension of $D$, the number of possible spatial derivative terms satisfies the following recursive formula:

$$S_D(d) = d + 1, \quad D = 1$$

$$S_D(d) = \sum_{i=0}^{d} S_{D-1}(i), \quad D > 1$$

where $S_D(d)$ represents the total number of potential derivative terms in a PDE system with a spatial dimension of $D$ and a maximum derivative order of $d$.

It should be noted that these spatial partial derivatives are inherent to almost all PDE systems and can be estimated directly from observational data. Therefore, we do not introduce any additional specific prior knowledge.

### C.2    EXPERIMENTAL RESULTS FOR DR AND KS SYSTEMS

First, we consider the DR system with control variable $u$. The mathematical formulation can be found in Table 2 of the main text. Here, we consider $u(x, t) = \sin(2\pi x / L_{DR}) + u_2(t)$, and $u_2(t)$ is a sampling function from GRF. After training using the FNODEs method, the experimental results are shown in Fig. S1. It is evident that our method exhibits high training efficiency and satisfactory predictive accuracy.

Similarly, we conducted experiments in the KS system using the function form described in Table 2 of the main text. Here, we consider $u(x, t) = \sin(2\pi x / L_{KS}) + u_3(t)$, and $u_3(t)$ is a sampling function from GRF. After training, the experimental results are shown in Figure S2. It is evident that our method also exhibits excellent predictive performance in the KS system.

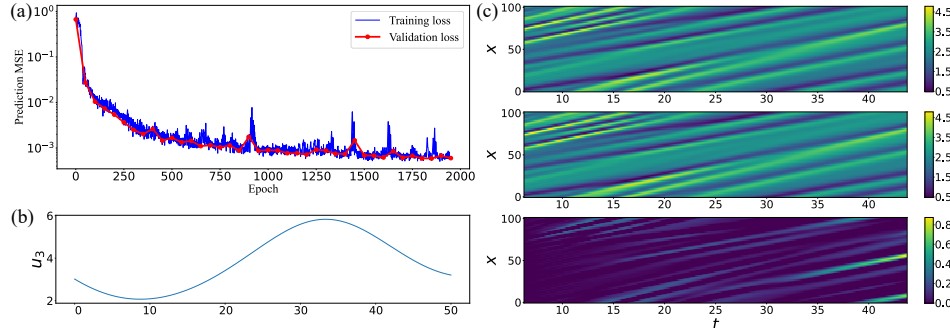

Figure S2: The experimental results of the KS system using FNODEs Method. (a) The variation of the training and validation loss during the training process. (b) Plot of $u_3(t)$ from a test data. (c) These three subplots from top to bottom represent the ground truth, predicted results, and prediction errors of test data.

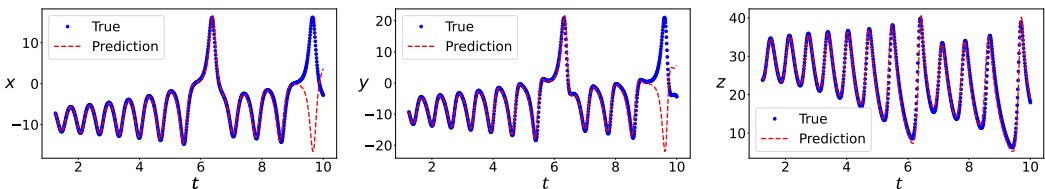

Figure S3: The prediction results of a test data in Lorenz system using FNODEs Method.

## C.3 EXPERIMENTAL RESULTS ON A CHAOTIC SYSTEM

Here, we consider a 3-D ODE system, known as the Lorenz63 system, which exhibits chaotic behavior. The system is described by the following equations:

$$
\begin{aligned}
\dot{x} &= 10(y - x), \\
\dot{y} &= \rho(t)x - y - xz, \\
\dot{z} &= xy - 8/3z,
\end{aligned}
$$

where $\rho(t)$ is a time-varying system parameter sampled from a Gaussian Random Field (GRF). Following the methods described in the main text, the training was completed in just 64.78 seconds for a total of 20,000 epochs. The experimental results are presented in Fig. S3. It is evident that our method demonstrates pretty good reconstruction performance on the chaotic Lorenz system.

## C.4 EXPERIMENTAL RESULTS ON A REAL-WORLD SYSTEM

To explore the potential applicability of the method presented in this paper in real-world systems, we conducted preliminary experiments in the time series data of polar motion (the data can be accessed via https://www.iers.org/IERS/EN/DataProducts/EarthOrientationData/eop.html). The polar motion components as the crucial Earth Orientation Parameters (EOPs), denoted usually by $x_p$ and $y_p$, describe the position of the Earth's instantaneous rotation axis with respect to the Earth's surface. In the field of geodesy, it is necessary to have accurate predictions of these parameters.

To enhance the accuracy of the dynamics modeling and prediction, we incorporate multiple distinct sources of physical information as features, following the guidelines of reference Shahvandi et al. (2022). Specifically, we utilize $dUT1 = UT1 - UTC$ and Length of Day (LOD) as the initial two features. Additionally, we also incorporate four Effective Angular Momentum (EAM) functions as features, namely: (a) Atmospheric Angular Momentum (AAM); (b) Hydrological Angular Momentum (HAM); (c) Oceanic Angular Momentum (OAM); and (d) Sea-Level Angular Momentum (SLAM) (the data can be accessed via http://rz-vm115.gfz-potsdam.de:8080/repository).

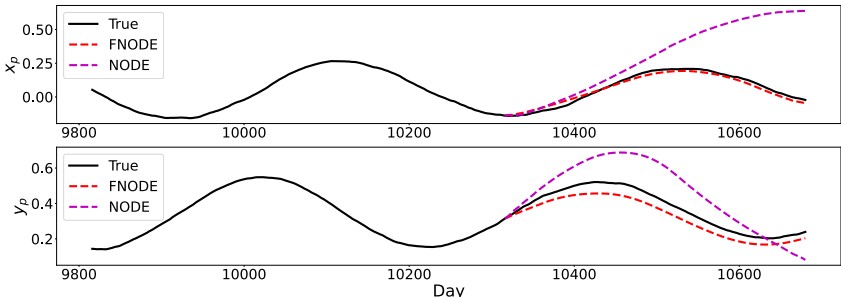

Figure S4: The prediction performance for a data segment using FNODE and NODE methods.

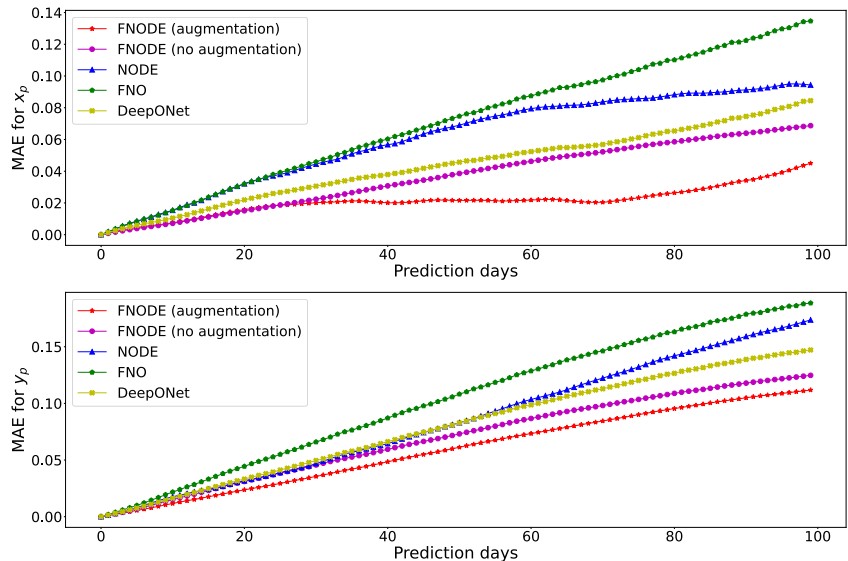

Figure S5: The prediction results for the polar motion data using different methods. Here, MAE refers to mean absolute error.

We partition the data as follows: 70% for training, 10% for validation, and the remaining 20% for testing. As shown in Fig. S4, our approach not only outperforms the traditional NODE method in prediction accuracy, but also reduces the training time by over an order of magnitude. Furthermore, we conducted a prediction for the subsequent 100 days from 50 randomly selected starting points under various methods, with the results depicted in Fig. S5. It is discernible that our method outperforms the baselines in both short-term and long-term predictions. Additionally, we also conduct a ablation experiment on data augmentation strategies (see FNODE (augmentation) and FNODE (no augmentation) in Fig. S5 for details). It is apparent from the results that incorporating feedback mechanisms during the training process can further enhance the predictive performance of the model.

## D   BASELINE METHODS

In this section, we provide a brief overview of the application of baseline methods in our experiments.

Firstly, in Section 4.2 of the main text, we utilize DeepONet as a baseline approach Lu et al. (2019). This method comprises a branch net and a trunk net. The branch net receives $m$ equidistantly sampled points from the parameter function $u(\boldsymbol{x}, t)$ as input and outputs a $p$-dimensional vector $\boldsymbol{b} = \{b_1, \cdots, b_p\}$. On the other hand, the trunk net takes the time $t$ and space $\boldsymbol{x}$ as input and outputs a $p$-dimensional vector $\boldsymbol{t} = \{t_1, \cdots, t_p\}$. Then, the system state of any input variable can

be predicted as follows:

$$\mathscr{G}(u)(\boldsymbol{x}, t) = \sum_{k=1}^{p} b_k(u) t_k(\boldsymbol{x}, t) + q,$$

where $q \in \mathbb{R}$ is a bias. After training, we can predict the state value $s(x, t)$ under any sampling function $u$.

Then, we consider the Fourier Neural Operator (FNO) Li et al. (2020) method, which formulates a neural operator by directly parameterizing the integral kernel in Fourier space. In practice, we take the system state $s(\boldsymbol{x}, t)$ and the parameter function $u(\boldsymbol{x}, t)$ at time $t$ as input and directly output the system state $s(\boldsymbol{x}, t + \delta_t)$ at time $t + \delta_t$. The FNO and DeepONet methods, as two neural operator approaches, effectively learn the prediction from the parameter function $u(t)$ to the system state. However, these methods lack interpretability and struggle to achieve better performance in extrapolation tasks beyond the distribution. Particularly, they exhibit poor performance in NS experiments where the initial values also vary.

Thirdly, we consider the PDE-NET Long et al. (2018) method, which leverages finite differences to approximate spatial derivative terms and uses a simple backward Euler for training and testing. In particular, for 2-d PDE systems, this method employs specific convolution kernels to compute spatial derivatives. In fact, this approach can be regarded as a numerical estimation of spatial partial derivatives and Euler iteration in the temporal direction. Although this is also an approach to estimating the gradient flow, in our experiments, the accuracy and speed of this method were inferior to the performance of the Fourier method.

Fourthly, we consider the NODEs for PDE systems. Indeed, due to the spatial derivatives of the PDE system, its dynamical behavior becomes highly intricate, posing numerous challenges in directly predicting the PDE system using NODEs. To execute the NODE method, we take the system state $s$ and the parameter $u$ as inputs to a neural network and obtain the temporal gradient of the system.

Fifthly, we contemplate utilizing the Physics-Informed Neural Networks (PINN) method Raissi et al. (2019) for the experiment. Unlike other methods, this approach does not train the neural network based on the prediction error of observational data. The method operates on the premise of known dynamical equations and employs a neural network to directly fit the initial conditions, boundary conditions, and dynamical equations. However, this approach cannot directly model a family of dynamical systems. Therefore, for each test data, the neural network must be retrained for dynamics prediction.

Sixthly, we consider the DeepAR method Flunkert et al. (2017), a forecasting method based on auto-regressive recurrent networks, which learns such a global model from historical data of all time series in the data set. However, our experimental results revealed that this method demonstrated poor performance in long-term prediction in differential equation systems, particularly in high-dimensional data within PDE systems. As a result, we only employ it as a baseline method for parametric ODE experiments.

Finally, we consider the message passing PDE solvers (MP-PDE) method Brandstetter et al. (2022). This method models the grid as a graph $\mathcal{G} = (\mathcal{V}, \mathcal{E})$ with nodes $i \in V$, edges $ij \in \mathcal{E}$, and node features $\boldsymbol{f}_i \in \mathbb{R}^c$. Here, the feature $\boldsymbol{f}$ consists of the spatial derivative terms for finite difference method (FDM), finite volume method (FVM), and WENO schemes. Then, we can update the graph representation through message passing, which in turn enables us to predict the future state of the system.

In spite of the effectiveness of the aforementioned methods in addressing parametric PDE problems, there is still room for improvement in terms of learning accuracy and extrapolation capabilities. To address this, we introduce the FNODEs method, which demonstrates superior performance compared to baseline methods across various experimental scenarios.

