# OpenReview forum: "From Fourier to Neural ODEs: Flow matching for modeling complex systems"
_ICLR.cc/2024/Conference — ICLR 2024 Conference Withdrawn Submission_

### Official Review · Reviewer_Xqgc · 2023-10-25

**Soundness:** 3 good
**Presentation:** 2 fair
**Contribution:** 3 good
**Rating:** 5
**Confidence:** 3

**Summary:**

In this paper, the authors propose a new method called Fourier NODEs (FNODEs). A key novelty of this work is that Fourier analysis is employed to estimate both temporal and spatial gradients of the noisy data. The estimated spatial gradients are fed into a neural network trained to estimate temporal gradients to assist the prediction of the temporal signals. In addition, the trained neural network could generate more data points through an ODE solver (like up-sampling). Comparisons with state-of-the-art methods showed efficacy of the proposed method regarding training time, accuracy and robustness.

**Strengths:**

1. The proposed method combines Fourier analysis with NODE and utilizes spatial gradients to improve temporal gradient estimations which looks novel.
2. By utilizing spatial gradients, it could up-sample training data and augment existing training data for model training (potentially improve model performance with more training data).
3. Experimental results look promising.

**Weaknesses:**

1. Novelty may be limited due to existing work. The authors may want to cite the following paper which also combines Fourier analysis with NODE and clarify their contributions: Hybrid Physical-Neural ODEs for Fast N-body Simulations.
2. Regarding architecture of the whole system, it’s not clear to me how the feedback loop works. For example, how the predicted data as feedback are combined with the observed data and used by the Fourier analysis? Why not encode the prediction error from ODESolver in the loss function of the neural network F_{\theta} as shown in the diagram of Fig. 1? The authors are encouraged to illustrate more on the motivations and methodology.

**Questions:**

1. In Sec. 1, the authors claim that Fourier analysis provides theoretical guarantees for accurately estimating the gradient flows of dynamical systems. Are there any citations to support this claim? Also are there any restrictions of the dynamical systems to make this claim work? e.g. continuity, differentiability and stochasticity of the dynamic system?
2. What’s the general guidance on selection of the cutoff in approximation of spatial gradients of PDEs. Same question goes to the control function u. For example, how to choose hyper parameters of the Gaussian random fields?
3. In the evaluation section, baseline methods are limited to ODE based methods. Would it make sense to compare it with state-of-the-art time series prediction methods like transformer, n-beats and deepar?

---

> ### Author Response · Authors · 2023-11-18
>
> We sincerely appreciate your insightful comments and valuable suggestions. We kindly recommend that the reviewer carefully reads the "**General response**". In response to the individual comments, we have thoroughly considered your suggestions and addressed all concerns in a point-by-point manner as outlined below. We hope that the reviewer recognizes the efforts and improvements we have made in both the methodology and experimental aspects.
>
> ```
> W1: Novelty may be limited due to existing work...
> ```
> **Response**:  Thank you for your suggestion, and we supplement this work as a reference. While both our work and theirs utilize Fourier analysis, our approaches differ fundamentally. Firstly, our study focuses primarily on a family of parametric ODE/PDE systems, which exhibit unique properties distinct from N-body simulations. Secondly, while they add the power spectral error to their loss function, our approach involves matching the gradient flow. Lastly, building on the powerful NODE framework, we propose an efficient and robust training strategy that is widely applicable, even extending to PDE systems.
>
> ```
> W2: Regarding architecture of the whole system, it’s not clear to me how the feedback loop works...
> ```
> **Response**:  Thank you for your careful reading and valuable comments. The feedback loop illustrated in Fig. 1 can be interpreted as a form of data augmentation, integrating the dynamics data predicted by the neural network with the original data. This strategy significantly expands the number of sample points, thereby enhancing the precision of Fourier analysis. For a more in-depth procedural understanding, please refer to Fig. 1 in the main text and Algorithm 1 in the appendix.
>
> Moreover, the use of gradient flow error as the loss function constitutes the focal point of our research. This choice is advocated due to the fact that in the classical NODE framework, utilizing prediction error as the loss function frequently leads to some challenges involving high computational costs and the propensity for finding local optima, rather than the global optimum.
>
> ```
> Q1: In Sec. 1, the authors claim that Fourier analysis provides theoretical guarantees...
> ```
> **Response**: Many thanks for your valuable comments. The theoretical guarantees of our method stem from Theorem 1 of the main text (proof in Appendix A.1), which provides conditions under which the gradient estimated via Fourier analysis consistently converges to the actual system gradient.
>
> It is noteworthy that our approach retains its efficacy even in scenarios where these conditions are not strictly met. For instance, many systems exhibit non-periodic behavior, and Fourier analysis, by default, assumes a periodic extension of the input discrete data. This may result in certain estimation errors at the boundaries. In practical implementation, we bypass this issue through disregarding the error at a small segment of the boundary and utilize the remaining error as the loss function (refer to code for details).
>
> Furthermore, as mentioned in the Point 1 of the "**General Response**", Fourier analysis and neural network complement each other by matching gradient flows from two distinct perspectives: spectral decomposition and dynamical modeling. In a statistical context, even with certain systematic errors in gradient estimation, this combination allows for robust dynamical modeling. Additionally, our experiments indicate that our method delivers satisfactory performance in modeling tasks for systems that do not strictly adhere to Theorem 1.
>
> ```
> Q2: What’s the general guidance on selection of the cutoff in approximation of spatial gradients of PDEs...
> ```
> **Response**: Thank you for your comment. In fact, our experimentation includes all common partial derivatives up to fifth order, without any prerequisite knowledge of dynamical equations. Through the neural network, our approach can automatically learn the complex interaction among the spatial derivatives, system state, and parameter values.
>
> As for the hyperparameters in GRF, similar to the articles on FNO and DeepONet methods, these hyperparameters are pre-established. And the variations within a reasonable range are acceptable for these parameters. Detailed configurations can be found in Appendix B and the code for data generation.
>
> ```
> Q3: In the evaluation section, baseline methods are limited...
> ```
> **Response**: Thanks for your instructive comment. In fact, our goal is to introduce a novel training strategy for NODE via flow matching, not to focus on achieving SOTA performances on various tasks compared with numerous baselines. Therefore, we employ the common baselines to validate the effectiveness of our proposed approach. For further reasons, please refer to the Point 3 of the "**General Response**".
>
> In addition, the methods like transformer, n-beats and deepar indeed perform well in some tasks of time series prediction, but it is not equipped to handle the parametric systems.

---

> > ### Author Response · Authors · 2023-11-23
> >
> > Dear Reviewer,
> >
> > Hope you are well! We would like to thank you for your constructive and insightful comments as well as your time and efforts you have dedicated to reviewing our work. We kindly remind you that the Author-Reviewer discussion is coming to an end and we would like to know if there are any remaining changes or questions you would like us to answer.
> >
> > In addition to the aforementioned revisions, we also supplement **three cutting-edge baseline methods**, namely **Physics-Informed Neural Networks (PINN)**, **Deep Auto-Regressive (DeepAR)**, and the **message-passing PDE solvers (MP-PDE)**. Additionally, we have supplemented the specific experimental results and methodologies to the revised article (please refer to the Illustrative experiments section and Appendix D in the revised manuscript). The experiments in the original manuscript and the additional real-world polar motion experiment, as well as the three newly added cutting-edge baseline methods, all sufficiently verify the efficiency and robustness of our method in modeling a wide range of complex systems. Therefore, our method exhibits SOTA performance in modeling tasks of the parametric ODE/PDE systems.
> >
> > Finally, we sincerely hope our significant revisions as well as our response can address your concerns, and we are more than happy to answer any further questions to make this work more readable and valuable. Many thanks again for your time, efforts, and valuable comments.
> >
> > Best,
> >
> > Authors

---

> > ### Comment · Reviewer_Xqgc · 2023-11-30
> >
> > Thanks for taking time to address the comments/questions. In general the responses make sense. I think if you can refine the paper based on comments from all reviewers, it will make an excellent paper. I am expecting more clear presentations/clarfications on contribution of this work and how it could push this research field forward. Right now I am more leaning towards to keep my original rating.

---

> ### Author Response · Authors · 2023-11-21
>
> Dear Reviewer,
>
> We hope this message finds you well! Thanks again for your valuable comments. We kindly remind you that the Author-Reviewer discussion is coming to an end and we would like to know if there are any remaining changes or questions you would like us to answer. We're more than happy to address these issues (before the rebuttal deadline). Otherwise, could you take this opportunity to re-evaluate our paper and update your score accordingly? Thanks!
>
> Best,
>
> Authors

---

### Official Review · Reviewer_533P · 2023-10-30

**Soundness:** 2 fair
**Presentation:** 1 poor
**Contribution:** 2 fair
**Rating:** 3
**Confidence:** 4

**Summary:**

The authors introduce a method to model forced time-dependent ODEs/PDEs. The method involves approximating the spatial and temporal derivatives with discrete Fourier transforms (DFTs) and use the applicable spatial derivatives as features to predict the temporal dynamics. The authors also use a data augmentation scheme to handle irregularly sampled data. Results are compared to baseline models for several common ODE/PDE benchmarks.

**Strengths:**

* Some level of novelty in using DFT to approximate derivatives and data augmentation to address data sparsity/irregularity.
* Experiments performed over a variety of systems.

**Weaknesses:**

* Clarity is really lacking - in general I feel that many important details are either explained in a confusing way or simply glossed over. I try to ask some of the questions below but overall they held me back from understanding the idea quite a bit.
* It is unclear how the model performs over longer periods of time, especially for the more complex benchmarks in KS and NS. This is what truly shows if the proposed model is competitive with existing methods.
* The baselines do not seem very competitive. The setup is not perfectly aligned but one should be able to adapt and compare baselines in [1].


References:

[1] Stachenfeld, Kimberly, et al. "Learned coarse models for efficient turbulence simulation." arXiv preprint arXiv:2112.15275 (2021).

**Questions:**

* How do you decide what spatial derivatives to pass as arguments to your model? How robust is the model if you do not get the terms exactly right?
* Does your training loss only involve predicting a single-step forward in time (based on equation 5)? Recent results (see [1] for example) suggest that using multiple steps improve performance significantly. This is also what the original NODE entails (computes multi-step error using continuous adjoint).
* For the data augmentation scheme, do you iteratively update the augmented data at every training step? The quality of the augmented data would obviously not be very good at the beginning of training. Do you take any special measures to account for such?
* Using DFT to approximate derivatives instead of finite difference obviously has its convenience but also comes with drawbacks. One of the more notable ones is the requirement that the underlying function should be smooth and periodic. It does not seem the periodic assumption is satisfied in your applications, which may lead to approximation errors (i.e. Gibbs phenomenon), especially at the boundaries. Do you use any strategies to address this?
* Are you only using DFT to compute derivatives but transform everything back in the real space when computing loss? This seems to be what's indicated in the text but then I see (page 6, fifth line from the bottom) reference to complex valued spatial derivatives in Fourier space.
* What is your exact definition of the error? Are you averaging everywhere (i.e. time, dimensions)? It would also be useful to show how the error accumulates as you roll out the model for longer times.
* How does your models do in terms of long-term error and stability?
* Figure 5 - equation (12) is not present in the text

References:

[1] Dmitrii Kochkov, Jamie A. Smith, Ayya Alieva, Qing Wang, Michael P. Brenner, and Stephan Hoyer. Machine learning–accelerated computational fluid dynamics. Proceedings of the National Academy of Sciences, 118 (21):e2101784118, 2021.

---

> ### Author Response · Authors · 2023-11-18
>
> Thank you for your valuable comments and helpful suggestions on the proposed framework. We kindly recommend that the reviewer reads the "**General response**" for a comprehensive overview of our main changes and clarifications.  We address each minor and major points raised in the review point by point as follows. We sincerely hope that the reviewer recognizes the efforts and improvements we have made, and reassess the presentation and contribution of our work.
>
> ```
> W1: Clarity is really lacking...
> ```
> **Response**:  Many thanks for your careful reading and helpful advice. We have conducted a comprehensive review and revision of the paper to enhance the readability and rigor. In what follows, we provide some explanations regarding the 8 detailed questions you care about.
>
> Q1: In fact, our experimentation includes spatial derivatives of up to fifth order, but without any prerequisite knowledge of dynamical equations. Leveraging the power of neural networks, our approach can robustly and automatically learn the complex interactions among the spatial derivatives, system state, and parameter values.
>
> Q2: Traditional NODE methods typically utilize the multi-step prediction error for training. However, our training strategy is different. As demonstrated in Eq. (5), we employ the Fourier analysis on a segment of training data to generate the temporal gradient estimate, denoted as $\bf{\hat{h}}’(t_n)$ at all sample points. The gradient errors at all points within this data segment then serve as the basis for our training loss.
>
> Q3: Prior to the implementation of the data augmentation scheme, we fully train the neural network using the existing data to acquire an optimal proxy model of the system dynamics. Subsequently, we further refine our model by training it with augmented data, thereby improving the accuracy of our modeling. The detailed implementation process can be found in the code file "multi_training.ipynb".
>
> Q4: In fact, many systems exhibit non-periodic behavior, and Fourier analysis, by default, assumes a periodic extension of the input discrete data. This assumption may introduce estimation errors, particularly at the boundaries. In practical implementation, we bypass this issue by disregarding the error at a small segment of the boundary and utilize the remaining error as the loss function to update neural network (refer to code for details). Furthermore, as mentioned in Point 1 of the "General Response," Fourier analysis and neural network complement each other by directly matching gradient flows from two distinct perspectives: spectral decomposition and dynamical modeling. In a statistical context, despite potential systematic errors in gradient estimation, this combination enables robust dynamical modeling.
>
> Q5: We apologize for any confusion in our earlier explanation. After computing the spatial derivatives in Fourier space, we proceed to transform them back to real space using the Inverse Discrete Fourier Transform. This conversion is essential before feeding the data into the neural network for the computation of temporal gradients.
>
> Q6: In the training phase, we consider the temporal gradient error in the response to Q2. When evaluating testing data, our metric involves calculating the mean prediction error across all sampling points. The accumulation of prediction errors over time is illustrated in Figs. 3-5 of the main text and Figs. S1-S3 in the appendix.
>
> Q7: Our experimental results, detailed in both the main text and the appendix, highlight the superior long-term stability of our method compared to the classical NODE. This enhanced stability is primarily attributed to our approach of directly incorporating the gradient error across the entire data segment into the loss function. This inclusion significantly improves the numerical integration stability of the neural network.
>
> Q8: We sincerely appreciate your comment and have addressed the identified error.
>
> ```
> W2: It is unclear how the model performs over longer periods...
> ```
> **Response**: Many thanks for your valuable comments. In fact, for complex systems, especially chaotic ones, almost all methods struggle to achieve long-term predictions. This is due to error accumulation through iterative predictions. Therefore, to assess various methods more effectively, appropriate extrapolation prediction times are set up for different systems.
>
> ```
> W3: The baselines do not seem very competitive...
> ```
> **Response**: Thanks for your instructive comment. In fact, our goal is to introduce a novel training strategy for NODE via flow matching, not to focus on achieving SOTA performances on various tasks compared with numerous baselines. Therefore, we employ the common baselines to validate the effectiveness of our approach. For further reasons, please refer to the Point 3 of the "**General Response**".
>
> In addition, the method from paper [1] does indeed perform well in the NS system, but it is not equipped to handle the parametric systems.

---

> ### Author Response · Authors · 2023-11-21
>
> Dear Reviewer,
>
> We hope this message finds you well! Thanks again for your valuable comments. We kindly remind you that the Author-Reviewer discussion is coming to an end and we would like to know if there are any remaining changes or questions you would like us to answer. We're more than happy to address these issues (before the rebuttal deadline). Otherwise, could you take this opportunity to re-evaluate our paper and update your score accordingly? Thanks!
>
> Best,
>
> Authors

---

> > ### Comment · Reviewer_533P · 2023-11-21
> > **Thank you for the response.**
> >
> > I want to thank the authors for the detailed response. After revision clarity has definitely improved. I still have reservations with regards to W2 and W3. In terms of W2, I agree that error accumulation is hard to handle, but it is extremely relevant for the method to be practically useful and precisely for this reason I would be more convinced to see where the proposed method stands with respect to the SOTA methods in this department. The arguments for W3 are also hard for me to buy in - since the authors are considering physics problem as the main selling point, NODE is not the most performant method and showing a strategy that improves upon that does not warrant significance in my opinion (i.e. one needs to show either an improvement on something NODE is SOTA or compare against SOTA for physics problems).

---

> > > ### Author Response · Authors · 2023-11-23
> > >
> > > Many thanks for your valuable response and further discussion. We acknowledge that comparing with state-of-the-art (SOTA) methods from recent years is a vital approach to verify the effectiveness of the proposed method. However, the selection of SOTA methods necessitates consideration of system characteristics, and even the same system may have different SOTA methods under different conditions.
> > >
> > > Considering that the subject of this research is parametric differential dynamic systems, our modeling and forecasting tasks should consider the property that the underlying dynamics are differential equations, but also consider the task scenario of modeling a family of dynamic systems. Consequently, the highly popular methods such as NODE, FNO, and DeepONet represent the SOTA baselines for the given task scenario. In addition,  our method could be regarded as a **gray-box** framework which incorporates the inductive bias from the dynamics perspective, that the underlying dynamics are governed by differential equation systems and all the potential spatial derivatives are fed into our model. Therefore, in comparison with several baselines, our framework exhibits better generalizations, more long-term and stable predictions, and requires fewer training samples.
> > >
> > > To verify the aforementioned discussion and further address the concerns of the reviewers regarding the baseline methods, we **supplement three cutting-edge baseline methods**, namely **Physics-Informed Neural Networks (PINN)**, **Deep Auto-Regressive (DeepAR)**, and the **message-passing PDE solvers (MP-PDE)**. Additionally, we have supplemented the specific experimental results and methodologies to the revised article (please refer to the Illustrative experiments section and Appendix D in the revised manuscript). Upon additional experiments, we find that the DeepAR method is more adept at handling the low-dimensional time series data with random noise, exhibiting greater error accumulation in our parametric ODE experiments. As for the parametric PDE experiments, It is observed that our proposed framework consistently yields lower prediction errors in comparison to the baseline methods. Here, it should be noted that our framework may require a longer prediction time compared to neural operator methods such as DeepONet and FNO, it exhibits superior performance in terms of the generalization, especially for NS systems with random initial values. Additionally, our framework surpasses the performance of the PDE-NET, thereby further validating the effectiveness of Fourier analysis-based estimation of spatiotemporal gradients over traditional numerical gradient estimation methods. For the baseline method of MP-PDE, as a black-box method in deep learning, it does not demonstrate satisfactory predictive performance in our experiments with a limited training set. In addition, the PINN method demonstrates performance similar to our approach, exhibiting robustness against fluctuations in training set size and noise levels. This is attributable to the necessity of employing underlying dynamical equations for training. Therefore, in data-driven modelling tasks (PINN is not applicable), our method indeed learned the underlying dynamics, demonstrating significant advantages in predictive capability.
> > >
> > > In conclusion, the experiments in the original manuscript and the additional real-world polar motion  experiment, as well as the three newly added cutting-edge baseline methods, all sufficiently verify the efficiency and robustness of our method in modeling a wide range of complex systems. Therefore, our method exhibits SOTA performance in modeling tasks of the parametric ODE/PDE systems.
> > >
> > > Finally, we sincerely hope our significant revisions as well as our response can address your concerns, and we are more than happy to answer any further questions to make this work more readable and valuable. Many thanks again for your time, efforts, and valuable comments.

---

### Official Review · Reviewer_PMhg · 2023-10-31

**Soundness:** 2 fair
**Presentation:** 3 good
**Contribution:** 1 poor
**Rating:** 3
**Confidence:** 3

**Summary:**

This paper introduced a way to improve the modeling of differential equations/dynamical systems with neural netwok. More precisely, it focuses on improving Neural Ordinary Differential Equation (Neural ODE), one of the popular frameworks in deep learning for dynamical systems in recent years. However, the training of Neural ODE is heavily computationally with the bottleneck in the backpropagation through nummerical ODE solver, and also often demonstrates undesired effects. To solve this, the authors of this work propose to incorporate Fourier analysis to approximate the ODE/PDE gradients, then use l2 loss to train this approximation with a parameterized neural network. The latter loss is taken from flow matching, a recent framework that shows promises in the generative modeling context. Evaluations on toy datasets show the gains in (decreased) training time and better MSE compared to Neural ODE.

**Strengths:**

The paper is in general based on solid theory and well-written.

**Weaknesses:**

* I have concern about the novelty of this paper: it is rather a combination of the flow matching framework for functional/time series data, with the closed-form velocity approximated by discrete Fourier transform.
* Since this leans on more methodological/empirical paper, I will comment more on the evaluation part. I do not think the authors have done a thorough literature survey. For example the related works/baselines comparison lack Physical Informed neural network [1], a rather popular framework that have performed some of the very similar tasks presented in the current paper. I am aware that for the modeling of time series/PDE. there are also score-based diffusion models that show competitive results, such as [2] and [3].
* To continue on the empirical evaluation, I do not understand why the authors did not include benchmarks on some of the realistic datasets, such as modeling time series. This is one of the main motivation of the paper, and I think stopping the evaluation at generated data in section 4.3 is inadequate.

[1] Raissi, Maziar, Paris Perdikaris, and George E. Karniadakis. "Physics-informed neural networks: A deep learning framework for solving forward and inverse problems involving nonlinear partial differential equations." Journal of Computational physics 378 (2019): 686-707.

[2] Li, Y., Lu, X., Wang, Y., & Dou, D. (2022). Generative time series forecasting with diffusion, denoise, and disentanglement. Advances in Neural Information Processing Systems, 35, 23009-23022.

[3] Apte, R., Nidhan, S., Ranade, R., & Pathak, J. (2023). Diffusion model based data generation for partial differential equations. arXiv preprint arXiv:2306.11075.

**Questions:**

See weaknesses.

---

> ### Author Response · Authors · 2023-11-18
>
> We thank you for your valuable comments and helpful suggestions on the proposed framework. We kindly recommend that the reviewer reads the "**General response**" for a comprehensive overview of our main changes and clarifications. Regarding the individual comments, we carefully consider your suggestions and address all the concerns point by point as follows. We sincerely hope that the reviewer recognizes the efforts and improvements we have made, and reassess the novelty and contribution of our work.
>
> ```
> W1: I have concern about the novelty of this paper: it is rather a combination of the flow matching framework for functional/time series data, with the closed-form velocity approximated by discrete Fourier transform.
> ```
> **Response**: Thank you for your comment. It is important to highlight that our approach should not be simplistically regarded as a fitting of the closed-form velocity. In fact, there is no definite closed-form Fourier function for the systems with variable parameters. Our approach, rooted in the NODE framework, introduces a novel strategy for effective and robust dynamical system modeling. In our framework, we leverage both Fourier analysis and neural networks synergistically, matching the gradient flows from two distinct perspectives: spectral decomposition and dynamical modeling. This integrated approach enhances the robustness of dynamic modeling a statistical sense. Additionally, we conducted comprehensive experiments across diverse systems, demonstrating consistently satisfactory performance in modeling tasks. For a thorough exploration and clarification of the novelty and contribution of our method, please refer to Point 1 in the "General Response".
>
> ```
> W2: Since this leans on more methodological/empirical paper, I will comment more on the evaluation part. I do not think the authors have done a thorough literature survey...
> ```
> **Response**: Thanks for your instructive comment. Indeed, there are numerous methods for modeling dynamical systems and predicting time series, each possessing distinct strengths and weaknesses in different application scenarios. In our work, our goal is to introduce a novel training strategy for NODE via flow matching, not to focus on achieving SOTA performances on various tasks compared with numerous baselines. Therefore, we employ the common baselines to validate the effectiveness of our proposed approach. For further reasons, please refer to the Point 3 of the "**General Response**".
>
> In addition, we did not consider the methods requiring additional system information or deep learning methods that rely on extensive training data as baselines. For example, the PINN method [1] necessitates the incorporation of dynamical equations for the computing the loss function, rendering it incompatible with the data-driven modeling objectives in our study. Regarding the score-based diffusion models, we acknowledge that these models, as purely data-driven black-box models, can exhibit commendable performance in certain time-series prediction tasks. However, our hypothesis that the underlying systems can be modeled by differential equations with time-varying parameters, and the proposed method and baselines considered in this study are more suited to these characteristics.
>
> Therefore, our proposed framework can be viewed as grey-box models, demanding less training data than deep black-box models, while concurrently offering enhanced the interpretability of the underlying dynamics.
>
> ```
> W3: To continue on the empirical evaluation, I do not understand why the authors did not include benchmarks on some of the realistic datasets...
> ```
>
> **Response**: Thanks for your careful reading and instructive advice. In this work, we primarily focus on introducing a novel approach for efficient and robust dynamical system modeling. In line with many method-centric investigations, such as DeepONet, PDE-Net, and some extensions of NODE, we primarily conduct preliminary validation on synthetic dataset. To further substantiate the potential applicability of our method in real data, we conducted an additional experiment on a real-world system. Detailed information can be found in the Point 2 of the "**General Response**" and Appendix C.4.

---

> ### Author Response · Authors · 2023-11-21
>
> Dear Reviewer,
>
> We hope this message finds you well! Thanks again for your valuable comments. We kindly remind you that the Author-Reviewer discussion is coming to an end and we would like to know if there are any remaining changes or questions you would like us to answer. We're more than happy to address these issues (before the rebuttal deadline). Otherwise, could you take this opportunity to re-evaluate our paper and update your score accordingly? Thanks!
>
> Best,
>
> Authors

---

> ### Comment · Reviewer_PMhg · 2023-11-22
> **Thank you for your rebuttal.**
>
> While I appreciate the efforts of the authors for improving their works and delivering the rebuttals, I still have major concern regarding the weak baselines. I also see that other reviewers agree with me on this point, and therefore decide to keep my current evaluation.

---

> > ### Author Response · Authors · 2023-11-23
> >
> > Many thanks for your valuable response and further discussion. We acknowledge that comparing with state-of-the-art (SOTA) methods from recent years is a vital approach to verify the effectiveness of the proposed method. However, the selection of SOTA methods necessitates consideration of system characteristics, and even the same system may have different SOTA methods under different conditions.
> >
> > Considering that the subject of this research is parametric differential dynamic systems, our modeling and forecasting tasks should consider the property that the underlying dynamics are differential equations, but also consider the task scenario of modeling a family of dynamic systems. Consequently, the highly popular methods such as NODE, FNO, and DeepONet represent the SOTA baselines for the given task scenario. In addition,  our method could be regarded as a **gray-box** framework which incorporates the inductive bias from the dynamics perspective, that the underlying dynamics are governed by differential equation systems and all the potential spatial derivatives are fed into our model. Therefore, in comparison with several baselines, our framework exhibits better generalizations, more long-term and stable predictions, and requires fewer training samples.
> >
> > To verify the aforementioned discussion and further address the concerns of the reviewers regarding the baseline methods, we **supplement three cutting-edge baseline methods**, namely **Physics-Informed Neural Networks (PINN)**, **Deep Auto-Regressive (DeepAR)**, and the **message-passing PDE solvers (MP-PDE)**. Additionally, we have supplemented the specific experimental results and methodologies to the revised article (please refer to the Illustrative experiments section and Appendix D in the revised manuscript). Upon additional experiments, we find that the DeepAR method is more adept at handling the low-dimensional time series data with random noise, exhibiting greater error accumulation in our parametric ODE experiments. As for the parametric PDE experiments, It is observed that our proposed framework consistently yields lower prediction errors in comparison to the baseline methods. Here, it should be noted that our framework may require a longer prediction time compared to neural operator methods such as DeepONet and FNO, it exhibits superior performance in terms of the generalization, especially for NS systems with random initial values. Additionally, our framework surpasses the performance of the PDE-NET, thereby further validating the effectiveness of Fourier analysis-based estimation of spatiotemporal gradients over traditional numerical gradient estimation methods. For the baseline method of MP-PDE, as a black-box method in deep learning, it does not demonstrate satisfactory predictive performance in our experiments with a limited training set. In addition, the PINN method demonstrates performance similar to our approach, exhibiting robustness against fluctuations in training set size and noise levels. This is attributable to the necessity of employing underlying dynamical equations for training. Therefore, in data-driven modelling tasks (PINN is not applicable), our method indeed learned the underlying dynamics, demonstrating significant advantages in predictive capability.
> >
> > In conclusion, the experiments in the original manuscript and the additional real-world polar motion  experiment, as well as the three newly added cutting-edge baseline methods, all sufficiently verify the efficiency and robustness of our method in modeling a wide range of complex systems. Therefore, our method exhibits SOTA performance in modeling tasks of the parametric ODE/PDE systems.
> >
> > Finally, we sincerely hope our significant revisions as well as our response can address your concerns, and we are more than happy to answer any further questions to make this work more readable and valuable. Many thanks again for your time, efforts, and valuable comments.

---

### Official Review · Reviewer_X9WB · 2023-11-01

**Soundness:** 3 good
**Presentation:** 3 good
**Contribution:** 3 good
**Rating:** 8
**Confidence:** 3

**Summary:**

The authors present a method that leverages flow matching loss for the learning of dynamical systems. Notably, the proposed algorithm does not require simulation, leading to a significant reduction in computational cost when modeling dynamical systems. The authors also introduce a novel augmentation strategy.

**Strengths:**

- The paper is well-written.
- The idea is simple and clear. It is supported by experimental results.

**Weaknesses:**

- Examples in the experimental part are a bit synthetic.

**Questions:**

1. Have you conducted an ablation study for the augmentation strategy?
2. What was the reason behind introducing the control functions? Does it simply add to the complexity of potential tasks?
3. Is the method applicable if the requirements in Theorem 1 are not satisfied?
4. Have you tried to apply this algorithm to real-life time series that are sampled from an unknown equation?

---

> ### Author Response · Authors · 2023-11-18
>
> We would like to thank the reviewer for the overall positive feedback and helpful suggestions. We revised the paper carefully according to the reviewer’s comments. Our major changes and clarifications are listed in “**General response**”.
>
> ```
> Q1: Have you conducted an ablation study for the augmentation strategy?
> ```
> **Response**:  Thank you for your comment. We have undertaken the ablation experiments as detailed in Section 4.3, and the comprehensive results are presented in Figure 5 of the main text. Notably, Figs. 5(b)-(e) demonstrate that the data augmentation training strategy significantly improves the modeling performance without requiring additional data samples. Furthermore, an ablation study on the newly incorporated real-world experiment has also been carried out, and further details can be found in Appendix C.4.
>
> ```
> Q2: What was the reason behind introducing the control functions? Does it simply add to the complexity of potential tasks?
> ```
> **Response**:  Many thanks for your valuable comment. In practical applications, it is essential to acknowledge that the underlying systems commonly involve time-varying control (external) parameters. Investigating such systems not only enhances the complexity of modeling tasks but also brings the modeling process into closer alignment with real-world scenarios. Of course, our method can be naturally applied to systems without control parameters.
>
> ```
> Q3: Is the method applicable if the requirements in Theorem 1 are not satisfied?
> ```
> **Response**: Thank you for your insightful comments. Indeed, when Theorem 1 is satisfied, the gradient estimation from Fourier analysis can consistently converge to the system's actual gradient, thereby enhancing the learning of underlying system dynamics. It is noteworthy that our approach maintains effectiveness even in scenarios where Theorem 1 is not strictly met. For instance, many systems exhibit non-periodic behavior, and Fourier analysis, by default, assumes a periodic extension of the input discrete data. This can result in certain estimation errors at the boundaries. In practical implementation, we disregard the error at a small segment of the boundary. The remaining errors are then utilized as the loss function to update the parameters of the neural network, a process detailed in the accompanying code.
>
> Furthermore, as highlighted in the Point 1 of the "**General Response**", Fourier analysis and neural network complement each other by directly matching the gradient flows from two distinct perspectives: spectral decomposition and dynamical modeling. In a statistical context, despite potential systematic errors in gradient estimation, this combination allows for robust dynamical modeling. Additionally, our experiments indicate that our method delivers satisfactory performance in modeling tasks for systems that do not strictly adhere to Theorem 1.
>
> ```
> Q4: Have you tried to apply this algorithm to real-life time series that are sampled from an unknown equation?
> ```
> **Response**: Thanks for your careful reading and instructive advice. In this work, we primarily focus on introducing a novel approach for efficient and robust dynamical system modeling. Aligning with numerous method-centric investigations, such as DeepONet, PDE-Net, and various extensions of NODE, we primarily conduct preliminary validation on synthetic data. To further substantiate the potential applicability of our method, we conducted an empirical experiment on a real-world system. Detailed information can be found in the Point 2 of the "General Response" and Appendix C.4.

---

### Author Response · Authors · 2023-11-18
**General response**

We would like to thank the reviewers for your time, efforts, and valuable comments and suggestions, which do help us to significantly improve the quality of this work. Accordingly, we try our best to enhance not only the methodology but also to supplement the additional experiments in order to clarify the contributions of our work. Specifically, the main changes and clarifications are listed as follows.

**1. The novelty and contribution of our work.**

The core contribution of our approach lies in a novel training strategy for flow matching, effectively **mitigates the issues of high computational costs and susceptibility to local optima** commonly encountered in traditional Neural ODE (NODE). To comprehensively grasp the intricacies of our method, the following elucidations are imperative.

Firstly, it is crucial to emphasize that our method should not be regarded merely as learning the closed-form velocity approximated by discrete Fourier transform. In our work, there is no definite closed-form Fourier function for systems with variable parameters. The Fourier analysis offers an estimation of the temporal gradient for each data segment from a spectral analysis perspective, serving as the target value for a neural network initiated from the dynamical perspective. Thus, from a statistical perspective, this gradient flow matching aims to facilitate faster and more robust dynamical modeling via neural networks, thereby achieving more accurate predictions.

Secondly, our framework can be extended to the resolution-invariant modeling of PDE systems. Here, Fourier analysis is employed to estimate the spatial gradients and potential high-order partial derivatives, which serve as additional inputs to the neural network. In contrast to Physics-Informed Neural Network (PINN) methods, our experimentation solely includes all common partial derivatives of up to the fifth order, but without embedding the any knowledge of the physical laws of the underlying systems.

Finally, it should be noted that a lot of existing deep learning methods for time series modeling are black boxes, which requires larger data samples to achieve a good result in a completely data-driven manner, such as the diffusion model, Transformer, n-beats mentioned by the reviewers. However, our method could be regarded as a gray-box framework which incorporates the inductive bias from the dynamics perspective, that the underlying dynamics are governed by differential equation systems and all the potential spatial derivatives are fed into our model. Thus, our framework exhibits better generalizations, fewer training samples compared to several baselines.

**2. Experiments on a real-world system**

In our study, we mainly focus on on refining the training strategy for NODE to enhance its efficiency and resilience. Following a routine akin to numerous method-centric investigations, we initially validate our approach using synthetic dataset. To further substantiate the potential applicability in real-world scenarios, we present preliminary experiments conducted on the polar motion dataset from from 1976 to 2023. The comprehensive details of our experimental setups and results are illustrated in **Appendix C.4**. The results indicate that our approach surpasses baseline methods in the prediction tasks, firmly establishing its substantial promise for practical applications.


**3. Selection of the baseline methods**

Indeed, there are numerous methods for dynamics modeling and time series prediction, each presenting the unique strengths and weaknesses tailored to specific application scenarios. NODE has been extensively utilized in prediction tasks across diverse systems, demonstrating its powerful capabilities for dynamics modeling. Our primary focus, as an extension of NODE, dose not lie in comparing the merits of the NODE framework against other state-of-the-art methods, as such comparisons have been thoroughly explored in the original and other various works. Instead, our aim is to provide a faster and more robust training strategy for NODE using a family of parametric systems, thereby enhancing the applicability and performance of the classic NODE framework.

Therefore, in our work, we consider the classic NODE method alongside well-established techniques like FNO, DeepONet, and PDE-Net, considering them as baselines. These baselines methods are particularly applicable to model parametric dynamical systems and have found extensive applications across various domains, thereby sufficiently substantiating the effectiveness of our approach.

**4. In addition, we have implemented several additional enhancements to augment the rigor and readability of the article. Please refer to the responses for each reviewer's comments.**

Finally, we thank all the reviewers again for your valuable and insightful comments. We hope that the revised article as well as the individual responses for each reviewer adequately addresses the reviewers’ concerns.